# Title page: SOIL Letters.

# Calcium is associated with specific soil organic carbon decomposition products.

Mike C. Rowley[1,2,3*], Jasquelin Pena[2,3*], Matthew A. Marcus[4], Rachel Porras[2], Elaine Pegoraro[2], Cyrill Zosso[1,5], Nicholas O. E. Ofiti[1,6], Guido L. B. Wiesenberg[1], Michael W. I. Schmidt[1], Margaret S. Torn[2,7], & Peter S. Nico[2,8*].

[1]Department of Geography, University of Zurich, Zurich, Switzerland.

[2]Earth and Environmental Sciences Area, Lawrence Berkeley National Laboratory, Berkeley, USA.

[3]Civil and Environmental Engineering, University of California, Davis, USA.

[4]Advanced Light Source, Lawrence Berkeley National Laboratory, Berkeley, USA.

[5]Climate and Agriculture, Agroscope, Zurich, Switzerland.

[6]Institute of Ecology and Evolution, University of Bern, Bern, Switzerland.

[7]Energy and Resources Group, University of California, Berkeley, USA.

[8]Department of Environmental Science, Policy, and Management, University of California, Berkeley, USA.

*Correspondence to*: *mike.rowley@geo.uzh.ch.

## Abstract

Calcium (Ca) may contribute to the preservation of soil organic carbon (SOC) in more ecosystems than previously thought. Here we provide evidence that Ca is co-located with SOC compounds that are enriched in aromatic and phenolic groups, across different acidic soil-types and locations with different ecosystem properties, differing in terms of climate, parent material, soil type, and vegetation. In turn, this co-localised fraction of Ca-SOC is removed through cation-exchange, and the association is then only re-established during decomposition in the presence of Ca (Ca addition incubation). Thus, highlighting a causative link between decomposition and the co-location of Ca with a characteristic fraction of SOC. Decomposition increases the

relative proportion of negatively charged functional groups, which can increase the propensity for the association between SOC and Ca, and in turn, this association inhibits dissolved organic carbon export or further decomposition. We propose that this mechanism could be driven by Ca hotspots on the microscale shifting local decomposition processes and thereby explaining the colocation of Ca with SOC of a specific composition across different acidic soil environments. Incorporating this biogeochemical process into Earth System Models could improve our understanding, predictions, and management of carbon dynamics in soils, and account for their response to Ca-rich amendments.

**Keywords**

Calcium addition, cation bridging, organo-mineral interactions, synchrotron-based spectromicroscopy, STXM C NEXAFS, Blodgett experimental forest, cation exchange, incubations.

**Non-technical summary**

This study shows calcium helps to preserve soil organic carbon in acidic soils, challenging previous beliefs that their interactions were largely limited to near-neutral to alkaline soils. Using spectromicroscopy, we found calcium is co-located with a specific type of carbon rich in aromatic and phenolic carbon. This association was disrupted when the calcium was removed and only reformed during decomposition with added calcium. This suggests that calcium amendments could enhance soil organic carbon stability.

**Competing interests**

The authors declare that they have no conflict of interest.

**Author contributions**

Mike C. Rowley[1,2,3*]: https://orcid.org/0000-0002-2440-7855
MCR was awarded the scholarship that part-funded the research, wrote the draft manuscript and beamline proposals, completed the experiments, data analysis, and figure creation.
Jasquelin Pena[2,3*] https://orcid.org/0000-0001-7081-3873
JP co-directed this research from its initiation and edited the initial fellowship proposal, beamline proposal, and manuscript numerous times.
Matthew A. Marcus[4]: https://orcid.org/0000-0003-2527-7586
MAM is the beamline scientist that runs the STXM C NEXAFS beamline (5.3.2.2). MAM edited the manuscript, helped with data creation and analysis, and wrote the program that was used to process the data.

Rachel Porras[2]: https://orcid.org/0000-0003-0251-100X

RP helped set up the incubation studies and edited the manuscript.

Elaine Pegoraro[2]: https://orcid.org/0000-0002-6865-8613

EP maintains the field site, helped to collect samples, lead parts of the bulk characterisation dataset collection, and edited the manuscript.

Cyrill Zosso[1,5]: https://orcid.org/0000-0002-7406-7908

CZ created parts of the bulk soil characterisation data and edited the manuscript.

Nicholas O. E. Ofiti[1,6]: https://orcid.org/0000-0003-3834-9040

NO helped create the bulk soil characterisation data and edited the manuscript.

Guido L. B. Wiesenberg[1]: https://orcid.org/0000-0003-2738-5775

GLBW directed the bulk characterisation data collection and edited the manuscript.

Michael W. I. Schmidt[1]: https://orcid.org/0000-0002-7227-0646

MWIS helped part-fund the project through the SNSF, directed the bulk data collection, and edited the manuscript.

Margaret S. Torn[2,5]: https://orcid.org/0000-0002-8174-0099

MST is the head of the belowground biogeochemistry scientific focus area team, manages the Blodgett Forest experiment, and

helped fund this research through the U.S. Department of Energy. MST also edited the initial fellowship proposal and manuscript.

Peter S. Nico[2,6*]: https://orcid.org/0000-0002-4180-9397

PSN co-directed this research from its initiation and edited the initial fellowship proposal, beamline proposal, and manuscript numerous times.


# 1.0 - Introduction

The accumulation and persistence of SOC is linked to its interactions with minerals and metal ions such as aluminium, iron (Fe), and calcium (Ca; Rasmussen et al., 2018; Kleber et al., 2021). The classical model of Ca-SOC bonding involves outer-sphere cation bridging ($Ca^{2+}$), where Ca bridges the negatively-charged surface of a clay mineral to a SOC carboxylic functional group (Edwards and Bremner, 1967; Oades, 1988). Yet, recent studies suggest that a wider range of interactions exist between Ca and SOC, driven by an array of interacting abiotic and biotic processes (Shabtai et al., 2023; Rowley et al., 2021; Beauvois et al., 2020). Specifically, Ca can abiotically influence SOC accumulation through its effects on soil aggregation and occlusion, or through different sorption processes involving various minerals or organic compounds (Fernández-Ugalde et al., 2014; Sowers et al., 2018). However, Ca can also play an important role in decomposition of SOC and has strong effects on microbial community composition and C use efficiency (Sridhar et al., 2022b; Sridhar et al., 2022a; Schroeder et al., 2024). For instance, Shabtai et al. (2023) demonstrated that Ca addition ($CaCl_2$) in mesocosms shifted the microbial community towards surface-colonising organisms, which enhanced C use efficiency, and decreased C mineralisation. Advanced fine-scale spectromicroscopy combined with targeted experiments would now offer a powerful approach to further unravel the relevance and interplay of these mechanisms across different soil environments.

Until recently, the effect of Ca on SOC was largely thought to be limited to soils with near-neutral to alkaline pH (Rasmussen et al., 2018; Rowley et al., 2018), or soils amended with alkaline minerals such as Ca carbonate (Paradelo et al., 2015). Yet, SOC is also co-located with Ca in carbonate-free, acidic soils as confirmed recently using scanning transmission X-ray microscopy coupled with carbon (C) near-edge X-ray absorption fine structure spectroscopy (STXM C NEXAFS; Rowley et al., 2023). In the acidic grassland soils at Point Reyes, California (hereafter Grassland), Ca was co-located with SOC that contained higher proportions of aromatic- and phenolic-C, and less O-alkyl-C, relative to the SOC associated with Fe. If identified in other acidic soil environments, this observation could challenge our conceptual understanding that the interactions between Ca and SOC are only limited to a narrow pH range in soils (Rowley et al., 2018; Rasmussen et al., 2018). We hypothesise that the co-location between Ca and a fraction of SOC rich in aromatic and phenolic functional groups in acidic grassland soils (Rowley et al., 2023) may be partly driven by microbial processes rather than physical or chemical processes alone. To confirm this hypothesis, additional STXM C NEXAFS measurements would be required on both natural samples from another acidic site with different ecosystem properties, and on samples subjected to experimental treatments, including Ca removal, addition, and decomposition in the presence or absence of Ca.

To test the mechanism(s) underlying the association of Ca with a specific fraction of SOC, we characterised samples (plant, litter, and soil) from the Blodgett Experimental Forest (hereafter Forest), Georgetown, California, using STXM C NEXAFS and bulk chemical techniques. We then monitored the response of soils to different experiments, including cation-exchange

and incubation, and compared the results to existing data from the Grassland (Rowley et al., 2023). The Forest site represents a temperate mixed-conifer site ecosystem, distinctly different from the Grassland in terms of climate, parent material, soil type, and vegetation (detailed in the Methods appendix). Notably the Forest has significantly less total Ca than the Grassland (> 20 cm depth; Fig. S1 & S2; Table S1), thereby enabling us to investigate whether the availability of Ca also influences its interaction with a specific fraction of SOC in acidic soils under different environmental conditions.

## 2.0 - Results and discussion

The STXM C K-edge NEXAFS spectra collected from acidic forest soils (pH = 3.7-6.2) show that SOC had a higher proportion of aromatic and phenolic C when co-located with Ca than with Fe (Fig. 1A & C). The C spectra of plant and litter samples from the Forest were similar irrespective of its co-location with Ca (Fig. S5); thereby, highlighting that the Ca-SOC association is not inherited from the inherent composition of plant or litter, but instead, seems to form in the soil. The characteristic fraction of SOC co-located with Ca in the Forest soils had a similar spectrum to that observed in the Grassland soils (Fig. S6; pH = 3.8-5.3; Rowley et al., 2023). Even though there was a large difference in total Ca content between the sites (Fig. S1 & S2; Table S1), the average (total) C spectrum was only slightly closer to the Fe-C spectrum in the Forest (linear combination fitting results = 83 % Fe-C *vs.* 17 % Ca-C) than at the Grassland (77 % Fe-C *vs.* 23 % Ca-C). Cluster and non-negative matrix factorisation (NNF) analysis of the C in organo-mineral assemblages (Fig. 1B) revealed that SOC was clustered into statistically relevant groups, which were strongly associated to the distribution of Fe and Ca (Fig. 1A & B; Fig. S7). Thus, across the samples that we investigated, Ca and Fe were co-located with statistically distinct fractions of SOC, implying that these elemental associations were important in dictating the distribution of SOC at the microscale (or *vice-versa*).

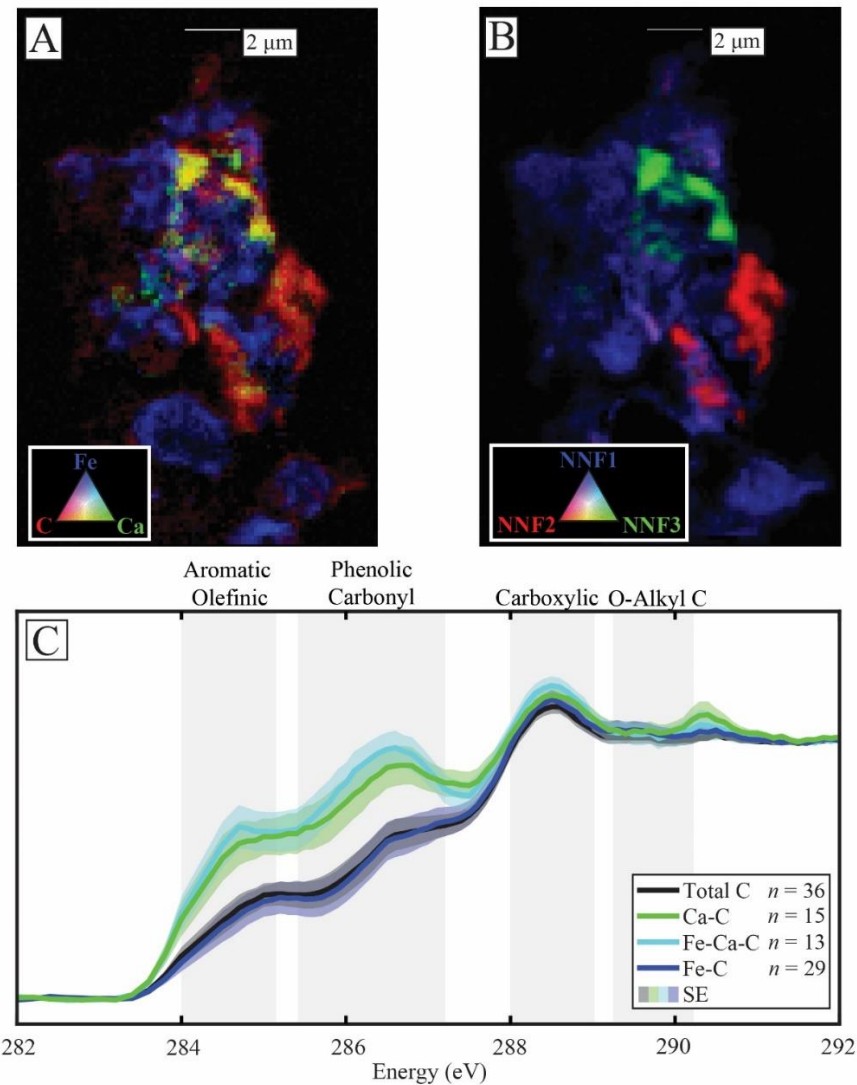

120

**Fig. 1. The microscale co-location of Ca or iron with specific carbon compounds in soils. A.)** A tricolour elemental map of the co-location of iron (blue), Ca (green), and carbon (red) in an organo-mineral assemblage from the 10-20 cm depth interval. **B.)** The fitting of non-negative matrix (NNF) factorisation statistical endmembers of carbon into clusters that correspond closely with its elemental distribution in A. The C (K-edge, $C_{1s}$) near-edge X-ray absorption structure spectra (NEXAFS) from these endmembers can be found in Fig. S7, in total there were 4 endmembers, but one was associated with a background signal. **C.)** The $C_{1s}$ NEXAFS spectra of the overall carbon (total C), carbon specifically associated with only Ca (Ca-C), only iron (Fe-C), or both Ca and iron (Fe-Ca-C), with the standard error of the averaged result fitted as a shaded area outside of the spectra. Regions of the spectra associated with specific functional groups are plotted in grey behind the spectra and were attained from Lehmann et al. (2009).

To test the mechanisms governing the co-location of Ca with a characteristic fraction of SOC, we conducted two experiments. First, we conducted a potassium cation-exchange (KCl) experiment to remove the Ca from soil samples and investigate its influence on SOC composition with STXM C NEXAFS. Cation-exchange leached the Ca and SOC co-located with it from our samples (Fig. 2A; Fig. S8 & & S9A), lowering the aromatic and phenolic C content, but also the aliphatic C content (*ca.* 287 eV). This experiment confirmed that Ca preserves a characteristic fraction of SOC and were consistent with reports that it can inhibit dissolved organic C (DOC) leaching in forest soils (Minick et al., 2017).

After this cation-exchange reaction, we re-introduced Ca to the samples, causing it to associate with soil particles and the SOC remaining after the KCl cation-exchange (Fig. 2A; Fig. S8). The STXM C NEXAFS spectra of samples post-exchange and Ca addition were similar to the Fe-C spectrum, with less aromatic and phenolic C than a typical Ca-C spectrum (Fig. 1C & 2A; Fig. S9). The C pool associated with Fe was resistant to this cation-exchange procedure, suggesting that it was probably bound by inner-sphere ligand-exchange reactions. Furthermore, the composition of the C co-located with Fe supports the hypothesis that Fe oxides preferentially bind microbially-transformed SOC, rather than new plant inputs (Fig. 1C & 2A; Spielvogel et al., 2008). Contrastingly, the co-location of Ca with a characteristic fraction of SOC was irreversibly disrupted by cation-exchange and, within the timeline of our experiments, could not be re-established through Ca addition alone (Fig. 2A; Fig. 7A).

In the second experiment, we incubated freshly collected soil samples after the addition of water, KCl as a control, or $CaCl_2$ (Fig. S4). Ca addition always reduced C mineralisation in our incubation experiments relative to the incubation with water (no KCl or $CaCl_2$; Fig. S4; Table S2). However, in our short-term experiments, unlike Shabtai et al. (2023), we did not see a significant decrease in C mineralisation relative to the monovalent cation control (0.2 M KCl; Fig. S4; Table S2). Yet, contrasting the results of the cation-exchange experiment (Fig. 2A), one month of microbial decomposition with added Ca shifted the STXM C NEXAFS spectra of total C towards the spectra associated with Ca-C co-location (Fig. 2B; Fig. S9B). Consequently, samples that were incubated with Ca had higher aromatic and phenolic, and lower O-alkyl C, in both the Ca-C and total C spectra relative to the pre-incubation spectra (Fig. S9B). This change in the STXM C NEXAFS spectra reproduced the observations in unaltered soil samples from the Forest and Grassland (Fig. 2B; Fig. S6 & Fig. S9) suggesting it was unlikely to be an artefact of the $Cl^-$ addition during incubation. The decrease in O-alkyl C, indicative of labile carbohydrates, in the incubated samples instead likely resulted from microbial decomposition prior to the association of remaining fraction of SOC with Ca and its subsequent preservation. In other words, the addition of Ca during incubation led to the preservation of a characteristic fraction of SOC co-located with Ca and significantly altered overall C composition. In conclusion, the association of Ca with a characteristic fraction of SOC does not form through physicochemical mechanisms alone, but instead, arises from coupled biogeochemical processes involving microbial decomposition.

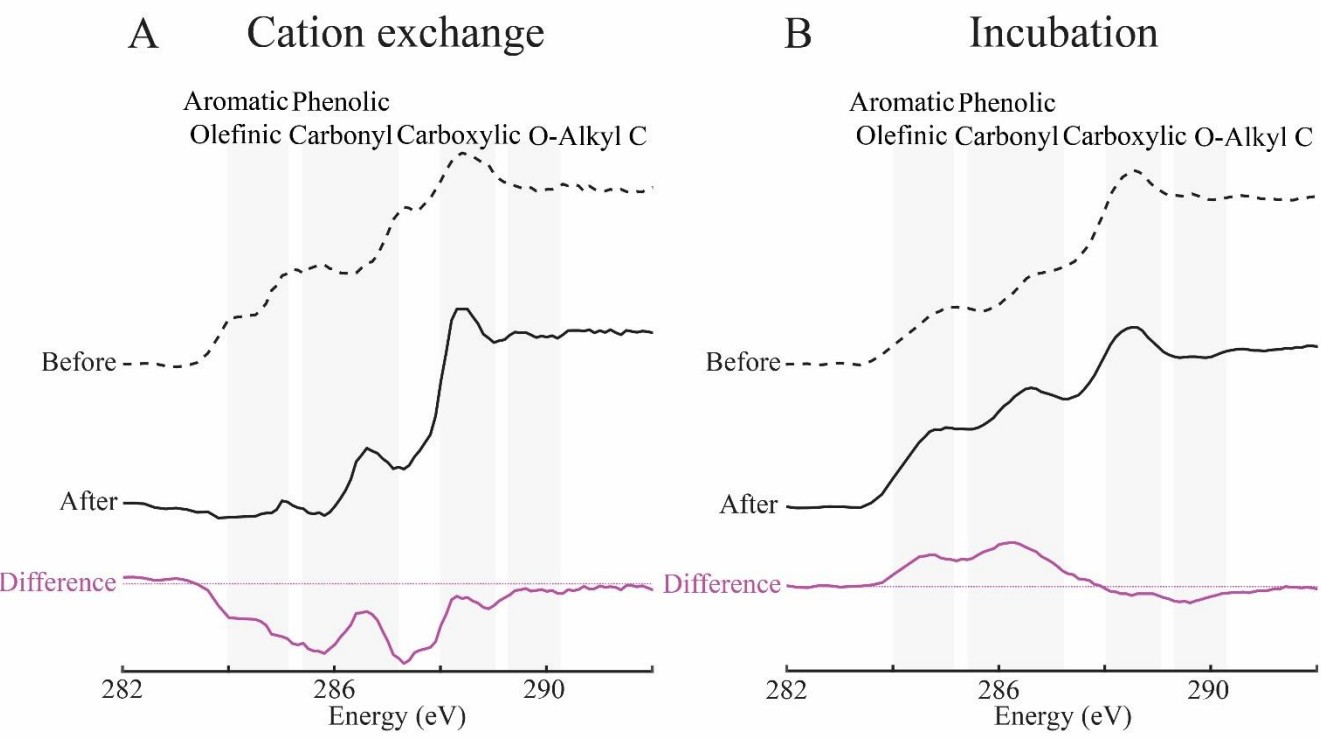

**Fig. 2. Targeted experiments demonstrate the importance of decomposition processes in the formation of the co-location of Ca with a characteristic fraction of soil organic carbon. A.)** The exchange of Ca with potassium (KCl) disrupted the carbon co-located with Ca (Fig. S8), reducing the associated aromatic and phenolic carbon in the $C_{1s}$ near-edge X-ray absorption fine structure (NEXAFS) spectra, and upon addition (CaCl$_2$), Ca re-associates with the remaining carbon (Fig. S8 & S9A). This remaining carbon has a spectrum like the carbon co-located with iron in unaltered samples (Fig. 1C), which was ultimately resistant to cation-exchange. **B.)** Incubation with added Ca reduced the O-alkyl carbon content in NEXAFS spectra while increasing the relative abundance of aromatic and phenolic carbon in the remaining sample (Fig. S9B). Thereby, microbial decomposition in the presence of added Ca reproduced a spectrum that was like the Ca-carbon spectrum of unaltered samples at the Grassland and Forest (Fig. 1C; Fig. S6). These experiments were run on two different samples from the same site explaining slight differences in the initial NEXAFS spectra, see methods for details.

The SOC fraction associated with Ca was enriched in aromatic and phenolic C, likely bound with Ca through carboxylic and phenolic functional groups in organo-mineral assemblages. Within the organo-mineral density fractions of calcareous soils, Grünewald et al. (2006) demonstrated a similar enrichment of lignin degradation products (rich in aromatic, phenolic, and carboxylic functional groups), attributing this to the preferential adsorption of either positively-charged layered hydroxide minerals (1.6-2.2 g cm$^{-3}$) or calcite (> 2.2 g cm$^{-3}$; Suzuki, 2002). As for acidic systems, Ca-binding affinities of C compounds

at pH 4.5 do indeed increase with the number of carboxylic groups (Tam and Mccoll, 1990). The relative proportion of negatively charged functional groups increases as SOC undergoes oxidative transformation or decomposition (Lehmann and Kleber, 2015; Lehmann et al., 2020), enhancing its propensity for Ca binding (Fig. S10). We thus propose a conceptual model in which decomposition by microbes is essential as the first step for the efficient formation of Ca-SOC association. In this scenario, initial decomposition and mineralisation of O-alkyl C increases the relative proportions of Ca-binding functional groups. Upon which, this decomposed fraction can then be bound by Ca, protecting it in organo-mineral assemblages, inhibiting its export as DOC, further decomposition, and ultimately its mineralisation, establishing its association with a characteristic fraction of SOC (Fig 2; Fig. S9).

The mechanism of co-location between Ca and a characteristic fraction of SOC was amplified by Ca addition (Fig. 2B; Shabtai et al., 2023; Sridhar et al., 2022a) and is consistently observed at the microscale across the different locations, depths, and acidic soil types. We can therefore further hypothesise that this mechanism of co-location is likely driven by Ca hotspots or abundance on the microscale, creating microdomains of decomposition that drive this characteristic association in natural samples (Kleber et al., 2021; Lehmann et al., 2020). Thus, a localised increase or microdomains in Ca availability could be driving local changes in microbial community (Shabtai et al., 2023; Sridevi et al., 2012; Sridhar et al., 2022a), decomposition pathways (Fig. 2B), the microscale distribution of C, and association of a characteristic fraction of SOC with Ca (Fig. 1C).

This study confirms that Ca is preferentially-associated with a characteristic fraction of SOC and the formation of this co-location is driven by chemically-altered decomposition processes. Uncovering this mechanism advances our understanding of SOC decomposition and its prediction in Earth System Models. Moreover, these mechanisms of SOC retention could be enhanced through Ca-rich agricultural amendments, such as those currently applied to acidic soils as part of agricultural and climate resilience practices, in particular liming or enhanced rock weathering (Shabtai et al., 2023; Paradelo et al., 2015; Xu et al., 2024; Vicca et al., 2022). To conclude, these data show that Ca plays key roles in the cascade of biogeochemical processes that affect SOC, its decomposition, and accumulation in more environments than previously thought.

**3.0 - Code/Data availability statement**

All data is freely available on ESS-DIVE (https://data.ess-dive.lbl.gov/). There is no code that is integral to the production of this manuscript.

**4.0 - Acknowledgements**

This research was based in part on material funded by the Swiss National Science Foundation (Grants P2LAP2_195077, P500PN_20665, & 200021_172744) and support to the Belowground Biogeochemistry Scientific Focus Area by the U.S. Department of Energy (DOE) Office of Science's Office of Biological and Environmental Research, Environmental Systems

Science Program (under contract number DE-AC02-05CH11231). This research used resources at the Advanced Light Source, a U.S. DOE Office of Science User Facility also under contract no. DE-AC02-05CH11231. We are grateful to beamline scientists and support staff from the Advanced Light Source, including David Shapiro, David Kilcoyne, and Andrea Jones. Special thanks to Stéphanie Grand, Patricia Fox, Robert Wagner, Yves Brügger, the Peña Lab, Belowground Biogeochemistry SFA team, Deep C project, and the E.S. team, for their support with various chemical analyses, enjoyable conversations, and tours.

## Appendix – Methods

Supplementary methods are also presented in the SI.

### Field site

Blodgett Experimental Forest (Forest) is situated in the Sierra Nevada foothills (1370 m asl) near Georgetown, California. The Forest soils were characterised as Alfisols, which are equivalent to Dystric Cambisols (Iuss Working Group Wrb, 2015), and had formed in granitic parent materials, in a temperate climate, under thinned, mixed-coniferous forest (Fig. S3; Gaudinski et al., 2009). The results from this paper are compared to soils sampled at Point Reyes (Grassland; Fig. S3), the full description of which can be found in Rowley et al. (2023). The Grassland soils were Luvisols or Lixisols developed in mixed sedimentary deposits, in a Mediterranean climate, under mixed-grassland species (Rowley et al., 2024), which spanned an acidic pH gradient (*ca.* soil $pH_{KCl}$ 4-5).

### Sampling

Samples were taken from the whole-soil warming experiment at the Forest. This section of the research centre has been subject to whole-soil warming of +4°C since the winter of 2013, the experimental details of which can be found in Hicks Pries et al. (2017). Soil cores were sampled from the control (soil cores 1-3) and warmed (soil cores 4-6) paired plots 1-3 in May 2021 (*n* = 6), while samples for the incubations, plant (leaf and branch), and litter samples were sampled in January 2023. Samples were oven dried at 40°C, sieved to 2 mm, with a separate, adjacent soil core sampled and sent to the University of Zurich for bulk characterisation (see SI for details).

### Bulk characterisation.

Soil pH was measured potentiometrically in 0.01 M Ca chloride ($CaCl_2$) solution at a 2:1 ratio with a Hamilton Polilyte Lab (238403) electrode. Total C and nitrogen contents were measured at the UC Davis Stable Isotope Facility using a Vario Micro Cube elemental analyser and Isoprime 100 isotope-ratio mass spectrometer. Total element contents were established using X-ray fluorescence (SPECTRO X-LAB 2000) without loss on ignition treatment.

**Experimental set-up**

To exchange Ca out from our samples and then add it back to the potassium-leached (exchanged) samples, we used the methods detailed in Whittinghill and Hobbie (2012). Briefly, we exchanged Ca from the soil core 4 (warmed plot 1) 60-70 cm sample using successive rinses with potassium chloride (KCl) solutions of decreasing strength (0.1 > 0.05 > 0.01 M KCl), before finally washing the sample with Milli-Q $H_2O$ (18.2 M$\Omega$ cm$^{-1}$ at 25°C). To add the Ca back into the exchanged samples we resuspended the samples with 0.1 M $CaCl_2$ (high Ca treatment equivalent) and then rinsed them again with Milli-Q $H_2O$. Samples were centrifuged between solutions, removing the supernatant, vortexed to resuspend samples in the subsequent rinse solution, before oven drying the remaining slurry on the final rinse with $H_2O$ at 40°C.

The incubation mesocosms were created by combining 20 g of surface soil samples from the Forest (0-20 cm) in sealed glass jars with either Milli-Q $H_2O$, 0.2 M KCl, or 0.1 M $CaCl_2$, equivalent to 20 c.mol$_c$. L$^{-1}$, and then incubated at 20°C for 1 month in the dark. Samples were maintained at 70 % field capacity throughout the incubation with Milli-Q $H_2O$. The respired $CO_2$ was sampled at different time points during the experiment, stored in pre-evacuated vials, replaced with Ultra Zero Air ($CO_2$ < 5 parts per million volume; Linde Gas and Equipment, Inc., Part # AI 0.0UZ-AS), and then measured using gas chromatography (Shimadzu). Air was regularly replaced with Ultra Zero Air to prevent toxic concentrations of $CO_2$ building up within the mesocosms, measuring and accounting for the removed $CO_2$. Initial SOC contents were used to calculate the cumulative C respired (mg $CO_2$-C respired g SOC$^{-1}$). Pre-testing (see SI for details) revealed that Ca addition always reduced C mineralisation in our incubation experiments relative to the incubation with water (no KCl or $CaCl_2$; Fig. S4; Table S2). However, in our short-term experiments, unlike Shabtai et al. (2023), we did not see a significant decrease in C mineralisation relative to the monovalent cation control (0.2 M KCl; Fig. S4; Table S2).

**STXM C NEXAFS**

We used STXM C NEXAFS to investigate the microscale physical and chemical association of SOC with Ca or Fe in samples using methods detailed in Rowley et al. (2023). Briefly, soil samples from 3 depth intervals (10-20, 40-50, and 60-70 cm) of soil cores 1-6 (control and warmed plots 1-3), plant (leaf and branch combined), litter samples from the field site, and samples from the exchange and incubation experiments were measured at beamline 5.3.2.2 of the Advanced Light Source. We combined observations from all plots ($n = 6$) as the exclusion of observations from the warmed plot samples (soil cores 4-6) had no significant effect on the STXM C NEXAFS spectra or our interpretations. Samples were spotted onto $Si_3N_4$ windows using methods adapted from Chen et al. (2014). Energy calibration was performed using $CO_2$ gas, setting the $1s{\rightarrow}3s\sigma_g$ peak in the C K-edge to 292.74 eV, and then checked at the end of the run (Prince et al., 1999).

**Statistical and data analysis**

All STXM C NEXAFS imaging and image analysis was completed in the STXM control program and C Fe STXM Image Reader (Marcus, 2023), respectively. A minimum of 2 STXM C NEXAFS image stacks were collected on each sample,

background subtracted for I$_0$, positionally aligned, and mapped for C (295-280 eV), Ca (394.4-342 eV), and Fe (710-698 eV). Image stacks were checked for saturation / thickness effects prior to further analysis. The image stacks were subset using a Boolean function in C Fe STXM Image Reader to isolate the C NEXAFS spectrum corresponding to the overall C, Ca associated C (no Fe), Fe-C (no Ca), or Fe-Ca-C signal (Rowley et al., 2023).

The STXM C NEXAFS stacks were investigated using principal component, clustering, and non-negative matrix factorisation analysis to group statistically relevant endmembers or standard spectra of clustered groups of C. These endmembers were then fit to the overall data using a least-squares fitting method. All exported spectra were background normalised in Athena (Ravel and Newville, 2005). The edge jump was set at 284.8 eV with an intensity of 1.0, the data were normalised by fitting a second-order polynomial to the post-edge region (291.8-302.0 eV) and the pre-edge was subtracted (279.8-283.3 eV; Rowley et al.,

2023). Linear combination fitting of the total C with Fe-C and Ca-C was completed in Athena (Ravel and Newville, 2005).

## 5.0 - References

Beauvois, A., Vantelon, D., Jestin, J., Rivard, C., Bouhnik-Le Coz, M., Dupont, A., Briois, V., Bizien, T., Sorrentino, A., Wu, B., Appavou, M.-S., Lotfi-Kalahroodi, E., Pierson-Wickmann, A.-C., and Davranche, M.: How does calcium drive the structural organization of iron–organic matter aggregates? A multiscale investigation, Environmental Science: Nano,
https://doi.org/10.1039/D0EN00412J, 2020.
Chen, C., Dynes, J. J., Wang, J., and Sparks, D. L.: Properties of Fe-organic matter associations via coprecipitation versus adsorption, Environmental Science & Technology, 48, 13751-13759, https://doi.org/10.1021/es503669u, 2014.
Edwards, A. P. and Bremner, J. M.: Microaggregates in soil, Journal of Soil Science, 18, 64, https://doi.org/10.1111/j.1365-2389.1967.tb01488.x, 1967.
Fernández-Ugalde, O., Virto, I., Barré, P., Apesteguía, M., Enrique, A., Imaz, M. J., and Bescansa, P.: Mechanisms of macroaggregate stabilisation by carbonates: implications for organic matter protection in semi-arid calcareous soils, Soil Research, 52, 180-192, https://doi.org/10.1071/SR13234, 2014.
Gaudinski, J. B., Torn, M. S., Riley, W. J., Swanston, C., Trumbore, S. E., Joslin, J. D., Majdi, H., Dawson, T. E., and Hanson, P. J.: Use of stored carbon reserves in growth of temperate tree roots and leaf buds: analyses using radiocarbon measurements
and modeling, Global Change Biology, 15, 992-1014, https://doi.org/10.1111/j.1365-2486.2008.01736.x, 2009.
Grünewald, G., Kaiser, K., Jahn, R., and Guggenberger, G.: Organic matter stabilization in young calcareous soils as revealed by density fractionation and analysis of lignin-derived constituents, Organic Geochemistry, 37, 1573-1589, https://doi.org/10.1016/j.orggeochem.2006.05.002, 2006.
Hicks Pries, C. E., Castanha, C., Porras, R. C., and Torn, M. S.: The whole-soil carbon flux in response to warming, Science,
355, 1420-1423, https://doi.org/10.1126/science.aal1319, 2017.
IUSS Working Group WRB: World reference base for soil resources 2014, update 2015, No 106, FAO, Rome2015.
Kleber, M., Bourg, I. C., Coward, E. K., Hansel, C. M., Myneni, S. C. B., and Nunan, N.: Dynamic interactions at the mineral–organic matter interface, Nature Reviews Earth & Environment, 2, 402-421, https://doi.org/10.1038/s43017-021-00162-y, 2021.
Lehmann, J. and Kleber, M.: The contentious nature of soil organic matter, Nature, 528, 60-68, https://doi.org/10.1038/nature16069, 2015.
Lehmann, J., Solomon, D., Brandes, J., Fleckenstein, H., Jacobsen, C., and Thieme, J.: Synchrotron-based near-edge X-ray spectroscopy of natural organic matter in soils and sediments, in: Biophysico-Chemical Processes Involving Natural Nonliving Organic Matter in Environmental Systems, edited by: Senesi, N., Xing, B., and Huang, P. M., John Wiley & Sons Inc.,
Hoboken, New Jersey, 723-775, https://doi.org/10.1002/9780470494950.ch17, 2009.

Lehmann, J., Hansel, C. M., Kaiser, C., Kleber, M., Maher, K., Manzoni, S., Nunan, N., Reichstein, M., Schimel, J. P., Torn, M. S., Wieder, W. R., and Kögel-Knabner, I.: Persistence of soil organic carbon caused by functional complexity, Nature Geoscience, 13, 529-534, https://doi.org/10.1038/s41561-020-0612-3, 2020.

Marcus, M. A.: Data analysis in spectroscopic STXM, Journal of Electron Spectroscopy and Related Phenomena, 264, 147310, https://doi.org/10.1016/j.elspec.2023.147310, 2023.

Minick, K. J., Fisk, M. C., and Groffman, P. M.: Soil Ca alters processes contributing to C and N retention in the Oa/A horizon of a northern hardwood forest, Biogeochemistry, 343-357, 10.1007/s10533-017-0307-z, 2017.

Oades, J. M.: The retention of organic matter in soils, Biogeochemistry, 5, 35-70, https://doi.org/10.1007/bf02180317, 1988.

Paradelo, R., Virto, I., and Chenu, C.: Net effect of liming on soil organic carbon stocks: A review, Agriculture, Ecosystems & Environment, 202, 98-107, https://doi.org/10.1016/j.agee.2015.01.005, 2015.

Prince, K. C., Avaldi, L., Coreno, M., Camilloni, R., and Simone, M. d.: Vibrational structure of core to Rydberg state excitations of carbon dioxide and dinitrogen oxide, Journal of Physics B: Atomic, Molecular and Optical Physics, 32, 2551-2567, https://doi.org/10.1088/0953-4075/32/11/307, 1999.

Rasmussen, C., Heckman, K., Wieder, W. R., Keiluweit, M., Lawrence, C. R., Berhe, A. A., Blankinship, J. C., Crow, S. E., Druhan, J. L., Hicks Pries, C. E., Marin-Spiotta, E., Plante, A. F., Schädel, C., Schimel, J. P., Sierra, C. A., Thompson, A., and Wagai, R.: Beyond clay: towards an improved set of variables for predicting soil organic matter content, Biogeochemistry, 137, 297-306, https://doi.org/10.1007/s10533-018-0424-3, 2018.

Ravel, B. and Newville, M.: Athena, Artemis, Hephaestus: Data analysis for X-ray absorption spectroscopy using IFEFFIT. [code], 2005.

Rowley, M. C., Grand, S., and Verrecchia, É. P.: Calcium-mediated stabilisation of soil organic carbon, Biogeochemistry, 137, 27-49, https://doi.org/10.1007/s10533-017-0410-1, 2018.

Rowley, M. C., Grand, S., Spangenberg, J. E., and Verrecchia, E. P.: Evidence linking calcium to increased organo-mineral association in soils, Biogeochemistry, 153, 223-241, https://doi.org/10.1007/s10533-021-00779-7, 2021.

Rowley, M. C., Falco, N., Pegoraro, E., Dafflon, B., Gerlein-Safdi, C., Wu, Y., Castanha, C., Peña, J., Nico, P. S., and Torn, M. S.: The importance of accounting for landscape position when investigating grasslands: A multidisciplinary characterisation of a California coastal grassland, Earth's Future, 12, 1-20, https://doi.org/10.1029/2023EF004208, 2024.

Rowley, M. C., Nico, P. S., Bone, S. E., Marcus, M. A., Pegoraro, E. F., Castanha, C., Kang, K., Bhattacharyya, A., Torn, M. S., and Peña, J.: Association between soil organic carbon and calcium in acidic grassland soils from Point Reyes National Seashore, CA, Biogeochemistry, 165, 91-111, https://doi.org/10.1007/s10533-023-01059-2, 2023.

Schroeder, J., Dămătîrcă, C., Bölscher, T., Chenu, C., Elsgaard, L., Tebbe, C. C., Skadell, L., and Poeplau, C.: Liming effects on microbial carbon use efficiency and its potential consequences for soil organic carbon stocks, Soil Biology and Biochemistry, 191, 1-20, https://doi.org/10.1016/j.soilbio.2024.109342, 2024.

Shabtai, I. A., Wilhelm, R. C., Schweizer, S. A., Höschen, C., Buckley, D. H., and Lehmann, J.: Calcium promotes persistent soil organic matter by altering microbial transformation of plant litter, Nature Communications, 14, 6609-6622, https://doi.org/10.1038/s41467-023-42291-6, 2023.

Sowers, T. D., Stuckey, J. W., and Sparks, D. L.: The synergistic effect of calcium on organic carbon sequestration to ferrihydrite, Geochemical Transactions, 19, 4, https://doi.org/10.1186/s12932-018-0049-4, 2018.

Spielvogel, S., Prietzel, J., and Kögel-Knabner, I.: Soil organic matter stabilization in acidic forest soils is preferential and soil type-specific, European Journal of Soil Science, 59, 674-692, https://doi.org/10.1111/j.1365-2389.2008.01030.x, 2008.

Sridevi, G., Minocha, R., Turlapati, S. A., Goldfarb, K. C., Brodie, E. L., Tisa, L. S., and Minocha, S. C.: Soil bacterial communities of a calcium-supplemented and a reference watershed at the Hubbard Brook Experimental Forest (HBEF), New Hampshire, USA, FEMS Microbiology Ecology, 79, 728-740, https://doi.org/10.1111/j.1574-6941.2011.01258.x, 2012.

Sridhar, B., Wilhelm, R. C., Debenport, S. J., Fahey, T. J., Buckley, D. H., and Goodale, C. L.: Microbial community shifts correspond with suppression of decomposition 25 years after liming of acidic forest soils, Global Change Biology, 28, 5399-5415, https://doi.org/10.1111/gcb.16321, 2022a.

Sridhar, B., Lawrence, G. B., Debenport, S. J., Fahey, T. J., Buckley, D. H., Wilhelm, R. C., and Goodale, C. L.: Watershed-scale liming reveals the short- and long-term effects of pH on the forest soil microbiome and carbon cycling, Environmental Microbiology, 24, 6184-6199, https://doi.org/10.1111/1462-2920.16119, 2022b.

Suzuki, S.: Black tea adsorption on calcium carbonate: A new application to chalk powder for brown powder materials,
335    Colloids and Surfaces A: Physicochemical and Engineering Aspects, 202, 81-91, http://dx.doi.org/10.1016/S0927-7757(01)01063-9, 2002.

Tam, S.-C. and McColl, J. G.: Aluminum- and calcium-binding affinities of some organic ligands in acidic conditions, Journal of Environmental Quality, 19, 514-520, https://doi.org/10.2134/jeq1990.00472425001900030027x, 1990.

Vicca, S., Goll, D. S., Hagens, M., Hartmann, J., Janssens, I. A., Neubeck, A., Peñuelas, J., Poblador, S., Rijnders, J., Sardans,
340    J., Struyf, E., Swoboda, P., van Groenigen, J. W., Vienne, A., and Verbruggen, E.: Is the climate change mitigation effect of enhanced silicate weathering governed by biological processes?, Global Change Biology, 28, 711-726, https://doi.org/10.1111/gcb.15993, 2022.

Whittinghill, K. A. and Hobbie, S. E.: Effects of pH and calcium on soil organic matter dynamics in Alaskan tundra, Biogeochemistry, 111, 569-581, https://doi.org/10.1007/s10533-011-9688-6, 2012.

345    Xu, T., Yuan, Z., Vicca, S., Goll, D. S., Li, G., Lin, L., Chen, H., Bi, B., Chen, Q., Li, C., Wang, X., Wang, C., Hao, Z., Fang, Y., and Beerling, D. J.: Enhanced silicate weathering accelerates forest carbon sequestration by stimulating the soil mineral carbon pump, Global Change Biology, 30, 1-17, https://doi.org/10.1111/gcb.17464, 2024.