# Peer review of "Calcium is associated with specific soil organic carbon decomposition products."

_EGUsphere, 2024_

## Author Response (AR1)

**Anonymous Reviewer 1**

The authors have put together a study that elucidates Ca-C interactions in acidic soils, the composition of C in these interactions, and the mechanisms involved. Overall, this manuscript constitutes an advancement in our understanding of SOC dynamics and our evolving understanding of the roles of calcium in SOC dynamics. It is within the scope of SOI, and quite relevant to its readership. The manuscript is short and well written, the introduction cites relevant articles and sets the reader up nicely to understand the context of the results. The figure are well made and nicely visualize the authors' findings. The main findings are nicely supported by the supplementary materials. I would have personally like for some of the table to be moved from SI to the main text to allow a more streamline reading experience. I recommend publishing this article after addressing the minor comments below.

**Dear Anonymous Reviewer 1,**

**We thank you for taking the time to review our article and thoroughly appreciate your constructive feedback. We have largely incorporated your feedback throughout and responded to your comments. Thank you again for making our article stronger.**

**While we agree that Table S1 could be a welcome addition to the main text, the SOIL Letters format is designed to be a shorter article format, which is why we didn't add a larger table in the main text.**

**Please find our responses to your specific minor comments below:**

Intro:

L39 - the associations between Ca and C in acidic soils is an interesting finding and indeed contrary to conventional thinking. However, if this is more prevalent than we assume, why do the concentrations of C and Ca not found to be important in acidic soils in data syntheses, e.g. Rasmussen et al 2018?

> **This is a very good point. As indicated in Fig. 3 of Rasmussen et al., 2018, it is instead likely the relative contribution of these interactions that increase with pH, as we move into soils with a higher proportion of Ca and as organic functional groups deprotonate. Yet there are many acidic to near-neutral environments that still have non-negligible concentrations of calcium, where calcium is cycled by vegetation into SOC rich horizons and where these interactions can still exist. As indicated by the linear-combination fitting results, it seems that it is just the relative importance that changes with pH, thus our results agree with Fig. 3 of Rasmussen et al., 2018 (we'll address this again in your next comment on the subject).**

L55 - how does the co-localization imply a microbial process? While microbial processes could very well be involved in many Ca-C interactions, no microbial measurements are presented here regarding the Rowley et al 2023. Could it not be solely the result of preferential associations between Ca and aromatic and phenolic C that form as OM decomposes in the presence of Ca? To me it makes sense to frame it such that is the hypothesis that requires testing, and to test it you performed the removal/addition/decomposition on several soils to evaluate whether it is a general feature.

**We agree with you and had already tried to word this as such (Speculatively / to confirm this hypothesis), while referencing the work of Shabtai et al., but we will make it clearer in the text:**

**"We hypothesise that the co-location between Ca and a characteristic fraction of SOC in acidic grassland soils (Rowley et al., 2023) may be partly driven by microbial processes rather than physical or chemical processes alone. To confirm this hypothesis, additional STXM C NEXAFS measurements are required on natural samples from another acidic site with different ecosystem properties, which we will subject to experimental treatments, including Ca removal, addition, and decomposition in the presence or absence of Ca."**

Also, could it also be the Ca that originated from the OM vs pedogenic Ca? Plant litter can contain significant amounts of Ca. Fig S5 shows agreement between Ca-C and total C in the litter and biomass, perhaps more carboxylic in Ca regions. Does this imply that Ca-C co localization at least partially results from decomposition of OM rich in Ca?

**This is another interesting point, also raised partially by anonymous reviewer 2. In both the Grassland (Rowley et al., 2023) and Forest (Fig. S5), we did not see the characteristic Ca-C STXM C NEXAFS spectra in colocalised Ca-C STXM measurements of litter and biomass. This thereby suggests that the co-localisation of Ca with a characteristic fraction of SOC (with a specific STXM C NEXAFS spectra) occurs due to a soil biogeochemical process. Yet, the origin of the Ca in this association could come from the surrounding soil or decomposing biomass, rich in Ca, and is likely a mixture of both.**

L56 - instead of characteristic fraction of SOC, perhaps state which fractions? (i.e. aromatic and phenolic rich)

**Here we will be more specific, but for brevity, we will continue to use characteristic fraction in later usages.**

**"the co-location between Ca and a characteristic fraction of SOC rich in aromatic and phenolic functional groups in acidic grassland soils"**

L65 - please elaborate on what you expect to find from a comparison of a soil with low Ca and a soil with high Ca.

**Great suggestion, we will expand on this specific sentence to make our reasoning clearer.**

**"Notably it has significantly less total Ca than the Grassland (> 20 cm depth; Fig. S1 & S2; Table S1), thereby enabling us to investigate whether variations in Ca content also influence the interaction of Ca with a specific fraction of SOC under different environmental conditions."**

Results:

L70 - are the litter and biomass C spectra shown in Fig S5 different than those of the Ca-C areas in the soil samples? If they are similar, one can pose a counterargument to whether this association is inherited from the composition of plant/litter.

**The litter and biomass STXM C NEXAFS spectra were collected on ground litter and biomass loaded onto Si$_3$N$_4$ windows while the soil sample STXM C NEXAFS spectra were collected on sieved soil loaded onto Si$_3$N$_4$ windows. The measurement and data processing are identical, but the initial samples are different. This thereby highlights that the difference in total C and Ca-C highlighted in Fig 1 C arises not due to the composition of aboveground biomass in the vegetation or litter, where there is largely no difference between Ca-C and total C, but instead belowground in the soil, through the mechanisms highlighted in the experiments (Fig. 2B).**

L74 - this is total Ca not exchangeable Ca, correct? It would make more sense to compare the exchangeable fraction, if that is the fraction that was removed and then re-added.

**While we have exchangeable Ca measurements for the Point Reyes sites, we do not currently have measurements for Blodgett Forest to compare them to. These measurements would be great to have in the future and plans have already been put forward to address this in a subsequent manuscript.**

L76 - does this mean C was roughly 77-83% Fe-associated and the rest Ca-associated? Suggesting Fe-C is quantitatively more important in these acidic soils? See my comment about the importance of Ca in acidic soils in the intro section.

**Yes, definitely, and this in turn agrees with our conventional understanding of SOC dynamics in these environments (Rasmussen et al., 2018). The Fe-C was also resistant to cation exchange as shown in Fig. 2A. However, it does act as a back-of-the-envelope calculation to suggest that approximately 20 % of the signal is associated in Ca-C in an environment where these interactions are usually overlooked. As per your comment above, it supports the idea that soil pH just dictates the relative importance of these associations in different soil environments (Rasmussen et al., 2018).**

L86 - it is unknown if DOC export has been prevented but perhaps that can be rephrased e.g. "consistent with reports of decreased DOC leaching..."

**Agreed and we will change it to "This experiment confirmed that Ca preserves a characteristic fraction of SOC, and is consistent with reports that it can decrease dissolved organic C (DOC) leaching in forest soils."**

L95 - why do you suppose addition of Ca to K-exchanged soils did not re-establish the original C-Ca interactions, while incubating fresh soils with Ca did. Could the K exchange process remove some of the POM that was a precursor to the aromatic/phenolic-C interacting with Ca? Other reasons?

**We hypothesise that the Ca addition after K exchange did not re-establish the original Ca-C interaction because these compounds had been leached from the samples. We also hypothesise that microbial decomposition processes are essential to the formation of this association. Thus, the association was not reformed in the timeline of our exchange experiments. The cation exchange likely leached a mixture of small particulate organic matter (nothing visible, but the occluded C content at Blodgett is typically low, average=10% of total C, range = 1-26%; Soong et al., 2021) stored within aggregates, disrupted by the cation exchange, and C associated with Ca as dissolved organic compounds. Thereby, removing these compounds from our soils, so when Ca was added**

**back to the samples it just associated with the remaining C (similar to Fe-C signal, which was resistant to cation exchange), but its typical relationship had been disrupted.**

L113 - the authors synthesize their results into a conceptual model that builds upon the continuum model (Lehmann and Kleber 2015) which is very useful. Their assumption is that Ca binds to C compounds mostly through carboxyl groups, which, as the authors state, are partially protonated at these pH values. Indeed, compounds often contain both phenolic and aromatic groups (lignin derivatives), however the hypothesis that microbial processing increases the carboxylic content thus supporting interactions with Ca is somewhat contrary to enrichment of Ca-C regions with aromatic and phenolic compounds. Could it be that Ca is interacting directly with phenolic and aromatic functional groups?

**This is another really good point. Yes, we believe that Ca is likely interacting with the negatively-charged phenolic and carboxylic functional groups of complex SOC biopolymers that are attached to a range of phenolic, aromatic, and carboxylic C groups. Ca-rich soils have indeed been shown to strongly stabilise lignin degradation products (Grünewald et al., 2006) and you'll also increase the relative proportions of phenolic groups in lignin degradation products during decomposition processes. It is possible that the carbon preferentially associated with Ca arises from lignin degradation products at Blodgett or Point Reyes, but this would require further experimentation with targeted analyses. We have expanded our hypothesis in the discussion section to include the Grunewald citation, which was removed during the editing process.**

**"The SOC fraction associated with Ca was enriched in aromatic and phenolic C, likely bound with Ca through carboxylic and phenolic functional groups in organo-mineral assemblages. Within organo-mineral density fractions of calcareous soils, Grünewald et al. (2006) demonstrated a similar enrichment of lignin degradation products (rich in aromatic, phenolic, and carboxylic functional groups), attributing this to the preferential adsorption of either positively-charged layered hydroxide minerals (1.6-2.2 g cm$^{-3}$) or calcite (>2.2 g cm$^{-3}$; Suzuki, 2002). As for acidic systems, Ca-binding affinities of C compounds at pH 4.5 do indeed increase with the number of carboxylic groups (Tam and Mccoll, 1990). The relative proportion of negatively charged functional groups increases as SOC undergoes oxidative transformation or decomposition (Lehmann and Kleber, 2015; Lehmann et al., 2020), enhancing its propensity for Ca binding (Fig. S10)."**

Fig 2A and Fig S9A - it appears that the exchange indeed removed phenolic C preferentially (~286 eV), but it also removed aliphatic C (~287 eV). Please address this point.

**This is an important point, thank you. We have made the decrease in aliphatic C clearer throughout, pointing to its removal during the K exchange experiment. Particularly on line 109. "Cation-exchange leached the Ca and SOC co-located with it from our samples (Fig. 2A; Fig. S8 & & S9A), lowering the aromatic and phenolic C content, but also the aliphatic C content (*ca.* 287 eV)."**

Methods:

The authors provide STXM images showing the effects of K exchange on removal of Ca, but I wonder if they measured Ca (and C!) in the rinse to attempt to quantify Ca-K exchange and how much C was removed through it.

**Not beyond Fig. S8, which clearly shows the removal of Ca from the samples and its replacement on the exchange complex with K in the elemental maps.**

Have the authors measured exchangeable Ca? This would seem to be more relevant to this study than total Ca.

**As indicated above, we do not have the exchangeable Ca measurements from Blodgett to compare them with the measurements of Point Reyes. This should eventually be measured in a future manuscript.**

**References**

Grünewald, G., Kaiser, K., Jahn, R., and Guggenberger, G.: Organic matter stabilization in young calcareous soils as revealed by density fractionation and analysis of lignin-derived constituents, Organic Geochemistry, 37, 1573-1589, 10.1016/j.orggeochem.2006.05.002, 2006.

Soong, J. L., Castanha, C., Hicks Pries, C. E., Ofiti, N., Porras, R. C., Riley, W. J., Schmidt, M. W. I., and Torn, M. S.: Five years of whole-soil warming led to loss of subsoil carbon stocks and increased CO2 efflux, Science Advances, 7, eabd1343, 10.1126/sciadv.abd1343, 2021.

**Anonymous Reviewer 2**

The manuscript provides novel mechanistic insights into a relevant soil process. After thorough inspection of the draft, my comments and recommendations mainly concern the presentation of the results, which should be well improved after significant revision related to the following points:

The manuscript is build up on a previous study and relates an earlier observation with some further experimental evidence. The presentation is kind of chronological which hinders the explanation unfolding topic-wise (e.g., in the first paragraph of the introduction the third sentence (l. 39) already jumps into fine-scale NEXAFS observation instead of elucidating the state of the art and the mechanisms further (and how fine-scale techniques provide particular insights based on the heterogeneous soil matrix), which I would recommend to strengthen the logical structure). Also the additional experimental steps (the washing) are inherently important to point out why these were done. This would improve reflecting on the following question: To which extent does the presented evidence suggest that the Ca is present at specific spots triggering an association with dissolved organic compounds or could the Ca be derived from organic compounds itself (and be inherited from the plant or microbial origin?). The microbial transformation is noted here and there but it does not become clear how it is specifically involved since specific aspects of the microbial role are less discussed (e.g., in l. 110 "arises from coupled biogeochemical processes involving microbial decomposition" or in l. 127&128 a lot of different microbial processes are noted but not interrelated with the actual observations). Maybe the authors should consider adding a graphical schema which briefly summarizes the specific details of the suggested mechanism, which I would leave up to them. However, I would suggest Fig. S10 is too speculative and simple since the molecular sizes were not measured. The explanation in l. 117-121 provides the key understanding and should be definitely elucidated further. Also, there is a lot of observations on the different mineralization rates found in the appendix methods and the SI, which is not well connected to the main body of the manuscript.

Dear Anonymous Reviewer 2,

We thank you for your review and detailed comments. Below I have included some of your comments from the summary above to help break apart our response. We thank you for your review, which has greatly improved the quality and clarity of our manuscript.

Below you will find responses to your specific comments:

The presentation is kind of chronological which hinders the explanation unfolding topic-wise…[ and ]…already jumps into fine-scale NEXAFS observation instead of elucidating the state of the art and the mechanisms further (and how fine-scale techniques provide particular insights based on the heterogeneous soil matrix), which I would recommend to strengthen the logical structure).

**Thank you for your comment, we have reorganized the structure of our introduction to match your suggestion. We have moved the second paragraph of the introduction above the first, with some slight modifications so that the state of the art and mechanisms come before the NEXAFS observations, thereby strengthening the logical structure of our introduction.**

Also the additional experimental steps (the washing) are inherently important to point out why these were done.

**Thank you, this was also raised by Anonymous Reviewer 1 and we have addressed it by highlighting our specific hypotheses and approach in the introduction. We have also adjusted line 90 to highlight the importance and reasoning behind of these experimental steps:**

**"First, we conducted a potassium cation-exchange (KCl) experiment to remove the Ca from soil samples and investigate its influence on SOC composition with STXM C NEXAFS"**

To which extent does the presented evidence suggest that the Ca is present at specific spots triggering an association with dissolved organic compounds or could the Ca be derived from organic compounds itself (and be inherited from the plant or microbial origin?).

**In Fig. S5 and also Rowley et al. (2023; see Fig. S17), where we measured aboveground biomass from Pt. Reyes with STXM C NEXAFS, there was no difference at either ecosystem between the total C and Ca-C signal. This demonstrates that the specific Ca-C association is unlikely to be inherited directly from plant biomass, such as protected particulate organic matter (cell wall debris for instance) rich in Ca, and instead arises due to a soil biogeochemical process. Due to the high aromatic and phenolic C content and reduced O-alkyl C content, we also believe it not to be directly of microbial origins. Instead, we suggest that this specific association arises during the decomposition of plant biomass by microorganisms, which is an essential step, the degradation products of which are then stabilized by Ca in mineral association, inhibiting its further decomposition or export as DOC. It would thus represent a mix of microbially-transformed, plant-like organic matter from both ecosystems.**

The microbial transformation is noted here and there but it does not become clear how it is specifically involved since specific aspects of the microbial role are less discussed (e.g., in l. 110 "arises from coupled biogeochemical processes involving microbial decomposition" or in l. 127&128 a lot of different microbial processes are noted but not interrelated with the actual observations). Maybe the authors should consider adding a graphical schema which briefly summarizes the specific details of the suggested mechanism, which I would leave up to them. However, I would suggest Fig. S10 is too speculative and simple since the molecular sizes were not measured. The explanation in l. 117-121 provides the key understanding and should be definitely elucidated further.

**If we had measured changes in the microbial community composition, we could speculate to the importance of specific communities, with specific strategies, or carbon use efficiencies (Shabtai et al., 2023), in creating the Ca-SOC colocalization. As we could not measure changes in the microbial community composition in this short-form study, we are limited in our ability to further speculate on the specific microbes and their mechanistic influence on SOC during decomposition. Although we hypothesize that the mechanism would be similar to those described in Shabtai et al., (2023), we have been cautious in our interpretations here.**

**Similarly with the conceptual model, we did design a more complex and complete conceptual model, but after much consideration, have remained conservative in our interpretations here and not included it. As highlighted by Anonymous Reviewer 1, we built upon the Lehmann and Kleber model to remain conservative, focusing instead on widely accepted soil chemical concepts and the role of Ca. Due to the spatial scale of our measurements (nm – 20 um), large particulate organic matter (> 250 um) will not be of much importance here. We thank you for your comments, but as we don't make a mention to their specific 600 Da cut-off, between large and small biopolymers; we posit that measurements of the molecular size of the C in Ca-C do not prevent us from making the link to an established SOC dynamics conceptual model (Lehmann and Kleber, 2015).**

Also, there is a lot of observations on the different mineralization rates found in the appendix methods and the SI, which is not well connected to the main body of the manuscript.

**Thank you for your comment. We would counter that these observations are essential to our narrative, as the incubation experiments are an important experimental element of the narrative. We are thus openly detailing the effect of Ca addition and our monovalent control on mineralization rates in the experiments in our SI.**

Further comments:

l. 1: The title could be misread as if Ca decomposes specific compounds. Also it might be advantageous to add that the results are related to acid soils.

**Thank you for your comment and raising the potential confusion in the title. After trialing this in many different ways, changing the title completely, and adding different components, we have decided not to edit the title. Our reasoning is that this article is about the association of Ca with a characteristic fraction of SOC and how this is formed, highlighting it arises through decomposition processes. We will also not add "in acidic soils" to the end of the title, as this is clearly highlighted in the rest of the document,**

including the first lines of the abstract. Thus, we request artistic license not to change the title, but address the rest of your comments below.

l. 19: What kind of different ecosystem properties?

**Thanks for highlighting this, we have added the specific details on the differences between the ecosystem properties of Point Reyes and Blodgett.**

**"Here we provide evidence that Ca is co-located with SOC compounds that are enriched in aromatic and phenolic groups, across different acidic soil-types and locations with different ecosystem properties, differing in terms of climate, parent material, soil type, and vegetation."**

l. 22: What kind of "conceptual and numerical models" are referred here? It might be better to directly describe the actual suggested mechanism and delineate it from other processes related with Ca. This comment is also related to l. 131 where it is not clear how "Earth System Models" are specifically influenced?

**This is a great comment that has clearly improved the quality of our abstract, thank you. We have used the remaining words, within the SOIL letters word limit for our abstract to break apart and clearly describe the actual suggested mechanism. We have also changed "conceptual and numerical models" to Earth System Models, which would be influenced by this mechanism if they accounted for the potential stabilization of SOC mediated by Ca and the influence of Ca-amendments.**

l. 98 It should be noted that a monovalent treatment does not seem to have been conducted in the reference cited here.

**The monovalent treatment is presented, but it is actually displayed in the supplementary information. You can find it in Fig. S1 of Shabtai et al., 2023.**

Fig. 1 and Fig. 2: It would be great of the panels had some subheadings, especially in Fig. 2. Also in Fig. 2 the x axis regions may need to be more clearly delineated.

**Thank you again for your comment. We have added subtitles to Fig. 2. While it is fully shown in the more detailed supplementary figure, and was requested by a coauthor, we have also added an X axis to Fig. 2. Thanks for your suggestion and improvements to our manuscript.**

**References**

Grünewald, G., Kaiser, K., Jahn, R., and Guggenberger, G.: Organic matter stabilization in young calcareous soils as revealed by density fractionation and analysis of lignin-derived constituents, Organic Geochemistry, 37, 1573-1589, https://doi.org/10.1016/j.orggeochem.2006.05.002, 2006.

Lehmann, J. and Kleber, M.: The contentious nature of soil organic matter, Nature, 528, 60-68, https://doi.org/10.1038/nature16069, 2015.

Lehmann, J., Hansel, C. M., Kaiser, C., Kleber, M., Maher, K., Manzoni, S., Nunan, N., Reichstein, M., Schimel, J. P., Torn, M. S., Wieder, W. R., and Kögel-Knabner, I.: Persistence

of soil organic carbon caused by functional complexity, Nature Geoscience, 13, 529-534, https://doi.org/10.1038/s41561-020-0612-3, 2020.

Rowley, M. C., Nico, P. S., Bone, S. E., Marcus, M. A., Pegoraro, E. F., Castanha, C., Kang, K., Bhattacharyya, A., Torn, M. S., and Peña, J.: Association between soil organic carbon and calcium in acidic grassland soils from Point Reyes National Seashore, CA, Biogeochemistry, 165, 91-111, https://doi.org/10.1007/s10533-023-01059-2, 2023.

Soong, J. L., Castanha, C., Hicks Pries, C. E., Ofiti, N., Porras, R. C., Riley, W. J., Schmidt, M. W. I., and Torn, M. S.: Five years of whole-soil warming led to loss of subsoil carbon stocks and increased CO2 efflux, Science Advances, 7, eabd1343, 10.1126/sciadv.abd1343, 2021.

Suzuki, S.: Black tea adsorption on calcium carbonate: A new application to chalk powder for brown powder materials, Colloids and Surfaces A: Physicochemical and Engineering Aspects, 202, 81-91, http://dx.doi.org/10.1016/S0927-7757(01)01063-9, 2002.

Tam, S.-C. and McColl, J. G.: Aluminum- and calcium-binding affinities of some organic ligands in acidic conditions, Journal of Environmental Quality, 19, 514-520, https://doi.org/10.2134/jeq1990.00472425001900030027x, 1990.